# A machine learning approach to support triaging of primary versus secondary headache patients using complete blood count

**Fei Yang**[1]*, **Tong Meng**[2], **Ben Torben-Nielsen**[1], **Carsten Magnus**[1], **Chuang Liu**[3], **Emilie Dejean**[4]

1 Roche Information Solutions, F. Hoffmann-La Roche AG, Basel, Switzerland, 2 Roche Molecular Systems, Santa Clara, California, United States of America, 3 Product Development Data and Statistical Sciences, Real World Data Enabling Platform, F. Hoffmann-La Roche AG, Basel, Switzerland, 4 Roche Diagnostics International Ltd, Rotkreuz, Switzerland

\* terry.yangfei@gmail.com

**Data Availability Statement:** This study is based in part on data from the Clinical Practice Research Database (CPRD) obtained under license from the

## Abstract

Headaches account for up to 4.5% of emergency department visits, where they present a significant diagnostic challenge. While primary headaches are benign, secondary headaches can be life-threatening. It is essential to rapidly differentiate between primary and secondary headaches as the latter require immediate diagnostic work-up. Current assessment relies on subjective measures; time constraints can result in overuse of diagnostic neuroimaging, prolonging diagnosis, and adding to economic burden. There is therefore an unmet need for a time- and cost-efficient, quantitative triaging tool to guide further diagnostic testing. Routine blood tests may provide important diagnostic and prognostic biomarkers indicating underlying headache causes. In this retrospective study (approved by the UK Medicines and Healthcare products Regulatory Agency Independent Scientific Advisory Committee for Clinical Practice Research Datalink (CPRD) research [20_000173]), UK CPRD real-world data from patients (n = 121,241) presenting with headache from 1993–2021 were used to generate a predictive model based on a machine learning (ML) approach for primary versus secondary headaches. A ML-based predictive model was constructed using two different methods (logistic regression and random forest) and the following predictors were evaluated: ten standard measurements of complete blood count (CBC) test, 19 ratios of the ten CBC test parameters, and patient demographic and clinical characteristics. The model's predictive performance was assessed using a set of cross-validated model performance metrics. The final predictive model showed modest predictive accuracy using the random forest method (balanced accuracy: 0.7405). The sensitivity, specificity, false negative rate (incorrect prediction of secondary headache as primary headache), and false positive rate (incorrect prediction of primary headache as secondary headache) were 58%, 90%, 10%, and 42%, respectively. The ML-based prediction model developed could provide a useful time- and cost-effective quantitative clinical tool to facilitate the triaging of patients presenting to the clinic with headache.

UK Medicines and Healthcare products Regulatory Agency (MHRA); however, the interpretation and conclusions contained in this report are those of the author(s) alone. The data used in this study are third-party data (i.e., data not owned or collected by the author(s)) provided by the CPRD under a license with the UK MHRA. A sub-license agreement/third-party agreement is required for data access by third parties (e.g., journal editors, reviewers, etc.) who were not included in the research team of the original study protocol approved by the CPRD's Research Data Governance process. The authors did not have any special access privileges that others would not have. Others can apply for access to the raw data (i.e., the data used to generate the relevant datasets for the current study) at the following URL: https://cprd.com/data-access/.

**Funding:** The study and third-party medical writing assistance was funded by Roche Diagnostics International Ltd (Rotkreuz, Switzerland). The funder (Roche Diagnostics International Ltd) through its employees at the time of the study (FY, TM, BT-N, CM, CL, and ED) had an active role in the design and conduct of the study; management, analysis, and interpretation of the data; preparation and review of the manuscript; and decision to submit the manuscript for publication.

**Competing interests:** I have read the journal's policy and the authors of this manuscript have the following competing interests: FY, TM, BT-N and CM are employees of Roche. CL and ED were employees of Roche at the time the study was conducted. This does not alter our adherence to PLOS ONE policies on sharing data and materials.

## Introduction

Headache is a common nervous system disorder, affecting approximately 50% of the general population [1, 2]. It can be classified as a primary headache disorder, typically migraine, tension, or cluster headaches [1], or a secondary headache disorder, which includes: giant cell arteritis, meningitis/encephalitis, subarachnoid hemorrhage (SAH), cerebral venous thrombosis, idiopathic intracranial hypertension, brain tumor, and ischemic stroke [1, 3, 4]. Although secondary headaches are rare compared with primary headaches, they are extremely important to recognize as they can require immediate intervention [1, 3], with further and rapid diagnostic evaluation including neuroimaging and lumbar puncture often needed [3, 5].

In primary care, headache is one of the most common presenting symptoms, with most due to primary causes, although many headache sufferers do not receive a specific diagnosis [6]. Headaches also account for up to 4.5% of all visits to the emergency department (ED) [2, 3, 5, 7, 8], with one observational study revealing an equal prevalence of primary versus secondary headache etiologies presenting to the ED (48 vs. 52%, respectively) [2]. Thus, accurate diagnosis of the underlying cause of headache and treatment initiation can be critical.

Headache has a historic reputation as being one of the most poorly classified neurologic disorders, and the International Headache Society classification system was developed to provide a hierarchy of criteria for diagnosis [9]. In the United Kingdom (UK) primary care setting, recognition of the importance of effective headache diagnosis, and management has prompted the additional training of general practitioners (GPs) with special interest in headache, and the establishment of headache clinics in general practice [10, 11]. Despite this, a varied degree of confidence in headache diagnosis and treatment still exists amongst emergency and primary clinicians, due in part to a lack of specific experience in neurology and poor use of the International Headache Society classification system [2, 6, 10]. A study conducted in Canada reported that misdiagnosis or diagnostic uncertainty occurred in more than one-third (35.7%) of cases of neurologic complaints in the ED, when comparing the initial diagnosis to the diagnosis made by the consulting neurologist [12]. In the UK, where there is a concerning shortfall of neurologists [10], the fear of clinical neurology by doctors outside the discipline has been described to amount to 'neurophobia' [10]. This can, together with patient anxiety and legal concerns, result in multiple appointments and unnecessary investigations, which may subsequently increase diagnosis times, healthcare costs, and economic burden [1, 4, 13].

The effective triaging of patients presenting with primary versus secondary headache is an important and currently unmet need [1, 5, 14]. Consideration of patients' medical history and physical examination are currently the most important aspects of headache assessment, and clinicians must be vigilant for "red flag" symptoms that are characteristic of serious secondary causes [1, 3]. This qualitative approach could be complimented by a quantitative methodologic tool, which could potentially streamline diagnosis, facilitate clinical decision-making, and reduce unnecessary investigations [2, 5, 14].

A complete blood count (CBC) is one of the most commonly requested blood tests [15] and its results may provide important diagnostic and prognostic biomarkers indicating underlying causes of headache. Several CBC parameters have been investigated for their ability to distinguish between primary and secondary headaches. A retrospective study of patients presenting to the ED with headache reported that leukocytosis or an increase in the percentage of polymorphonuclear leukocytes (PMNs) had a sensitivity of 89.8%, a specificity of 46.7%, a positive predictive value of 82.1%, and a negative predictive value of 62.8% for diagnosing SAH within 6 or 12 hours of ED admission [16]. The ratio of neutrophils to lymphocytes (NLR) has also received increasing attention as a diagnostic and prognostic marker of inflammation and can be easily calculated from standard measurements of CBC tests. A retrospective study found

NLR to be higher in people presenting to the ED with a migraine attack versus people without a headache [17]. A further retrospective, single-center analysis in ED patients presenting with headache accompanied by nausea and vomiting, found that NLR could distinguish between those with migraine and those with SAH [16]. Although this retrospective study involved a limited number of clinical cases, median NLR values were found to be significantly higher in patients with SAH compared with those with migraine and other headaches (both p < 0.001) [18].

Measurements derived from CBC tests have also been evaluated in the prediction of secondary headache severity. In one retrospective, single-center review, patients with SAH whose leukocyte count was >15 x $10^9$/L during admission were more than three times more likely to develop vasospasm [19]. In another study of patients with SAH, the authors reported that patients admitted with spontaneous SAH with a leukocyte count of >20,000 had a mortality rate of 50% [20]. Furthermore, mean platelet volume and platelet distribution width has been shown to be increased in patients with cerebral venous thrombosis and brain parenchymal lesions compared with patients with cerebral venous thrombosis without lesions [21].

Aligning with the concept of using routine blood test results as a triaging tool to assist physicians when deciding whether to perform neuroimaging on patients presenting with severe headache, this study was designed to evaluate a machine learning (ML)-based approach to classify primary versus secondary headache using real-world data (RWD) derived from a large patient group.

## Materials and methods

### Study design

This was a retrospective, observational study of RWD from patients presenting with a complaint of headache to the clinic. Due to the limitation of accessing RWD in the ED and evaluating the suitability to the study objectives, primary care data from the UK Clinical Practice Research Datalink (CPRD) were used.

### Database

The CPRD includes longitudinal electronic health records generated by GP practices in the UK. The current study included patients from two CPRD databases [22–24]: CPRD GOLD (using a data-cut of July 2021), and CPRD Aurum (using a data-cut of June 2021).

### Study population

Eligible patients with a record of presenting to a GP practice with a complaint of headache were identified in the CPRD GOLD and CPRD Aurum databases separately, according to the list of codes provided in S1–S10 Tables. The date of the patient's first clinic visit with headache symptoms was defined as the index date. Patients were included for analysis if they: 1) received a diagnosis of either primary or secondary headache within a 30-day window after the index date, 2) had laboratory results available for all ten specific parameters from the CBC test within a 30-day window after the index date and 3) had data classed as of "research acceptable" quality by the CPRD. Qualifying patients from both CPRD GOLD and CPRD Aurum were then merged to form an analytical dataset. To avoid possible duplicate individuals in the analytical dataset, the following steps were employed: 1) if a patient had multiple sets of records that fulfilled the inclusion criteria, only data from their latest index date were included; and 2) for clinics that changed enrolment from CPRD GOLD to CPRD Aurum, qualifying patients were

only identified during the period that the clinic was enrolled for CPRD Aurum and not for CPRD GOLD.

The study was conducted in accordance with the principles founded in the Declaration of Helsinki of 1975 (revised 2013). The study was approved by the UK Medicines and Healthcare products Regulatory Agency Independent Scientific Advisory Committee for CPRD research (20_000173). All patient data were anonymized; thus, the requirement for patient consent was waived. Individual patients can opt out of sharing their records with the CPRD. An overview of the study is available online [25].

### Outcome definitions and study variables

Primary headaches were defined as migraine, tension-type headache, and cluster headache, while secondary headaches were defined as those caused by ischemic stroke, cerebral venous thrombosis, hemorrhage (including SAH), arteritis, and angiitis (S1–S6 Tables). In the event that a patient had diagnoses contributing to both primary and secondary headaches during the same episode, the patient was categorized as having secondary headache.

Measured values of the following ten CBC parameters were included: red blood cell (RBC) count, platelet count, mean corpuscular volume (MCV), white blood cell (WBC) count, neutrophil count, lymphocyte count, monocyte count, eosinophil count, basophil count, and hemoglobin. The following 19 ratios of the individual parameters were also contrived as variables: platelet/RBC, WBC/RBC, RBC/neutrophil, RBC/monocyte, monocyte/eosinophil, platelet/MCV, platelet/lymphocyte, platelet/eosinophil, MCV/WBC, MCV/neutrophil, neutrophil/lymphocyte, neutrophil/eosinophil, lymphocyte/monocyte, lymphocyte/eosinophil, hemoglobin/lymphocyte, hemoglobin/eosinophil, hemoglobin/RBC, MCV/monocyte, and MCV/hemoglobin.

### Descriptive analysis

First, patients with primary headaches and secondary headaches included in the analytical cohort were described, and the differences between the two groups were assessed using t-test for continuous variables and chi-squared test for categorical variables. The Hotelling $T^2$-test was then used to examine whether data from the primary and secondary headache groups could be differentiated based on several collective variables, and was performed using the T2. test function from the R-project for statistical computing [26].

Data from patients with outlier results (the top and bottom 1% of the values) for any of the ten CBC parameters and body mass index, were excluded from the analysis.

### Prediction model development, performance, and validation

To develop the prediction model for primary headaches versus secondary headaches, two separate ML-based approaches (logistic regression and random forest) were used and evaluated. A total of 31 candidate predictors for the prediction models were preselected, including: demographic variables (age and sex), ten parameters from the CBC test (blood cell counts) and 19 variables derived from the ratios of the ten parameters from the CBC test. Earlier reports indicate that variables comprising specific blood cell ratios are medically relevant [17, 18].

Data normalization and optimization was performed by min–max scaling to standardize the scale of CBC-related values included in the model (preliminary tests showed that min–max scaling outperformed z-score scaling). No data balancing technique was used, rather, all available data in the analytical cohort were used.

## Model construction

The ability of the blood cell count ratios to predict headache type was determined by comparing the predictive performance metrics of each prediction model with and without the 19 variables derived from the ratios of the ten parameters from the CBC test. As part of the model specification, the following feature selection techniques were assessed with the aim of simplifying the model while maintaining good performance: lasso regularization [27], recursive feature elimination [28], weight of feature importance (analyzed by logistic regression), and correlation analysis for any candidate predictor with headache type.

For evaluation of the prediction models developed using logistic regression and random forest, five-fold cross-validation was used to obtain the mean and standard deviation for each of the following model performance metrics: 1) accuracy; 2) balanced accuracy; 3) average precision; 4) F1-score; and 5) area under the curve. Since the data were imbalanced and secondary headache is more serious than primary headache, the model was tuned by changing the predicted probability threshold cut off from 0.5 to 0.3. This was done with the aim of finding a prediction model that balanced the correct detection of secondary headaches with a false negative rate (i.e. incorrect prediction of secondary headache as primary headache) of less than 10%, whilst minimizing overcalling of primary headache as secondary headache. The final prediction model was then evaluated using a single 80:20 train/test dataset split. To visualize and summarize the performance of the prediction model, a confusion matrix was generated.

Standard descriptive statistical analyses were conducted using R. Machine learning software (R 4.1.3) [29], and ML analyses were implemented in Python (3.9.0), using the scikit-learn library [30, 31]. Matplotlib was used for visualization [32].

## Results

### Cohort description

A total of 121,241 patients satisfied the inclusion/exclusion criteria and formed the analytical cohort, comprising 108,906 patients with primary headache and 12,335 patients with secondary headache (Fig 1). Baseline demographic and clinical characteristics of the analytical cohort, stratified into primary and secondary headache groups, are presented in Table 1. Overall, patients presenting with primary headache were significantly younger than those presenting with secondary headache (mean 44 vs. 70 years; $p < 0.001$). The majority of primary headache patients were between 21 and 50 years of age (n = 64,004; 58.7%), whereas the majority of secondary headache patients were between 61 and 80 years of age (n = 6,454; 52.3%). In both headache groups, there were considerably more female than male patients (78.1% vs. 21.9% and 65.7% vs. 34.3%, for primary and secondary headache groups, respectively).

### Descriptive statistical analyses

Results from descriptive statistical analyses revealed statistically significant differences ($p < 0.001$) between the population means for almost all of the ten parameters from the CBC tests and the 19 ratio variables measured in patients between headache groups (Tables 2 and 3). However, there were substantial overlaps in the range of values for these variables with broadly similar mean values (S1 and S2 Figs). Further analysis using the Hotelling $T^2$-test showed that the two headache groups could not be properly differentiated (Table 4).

### Predicting primary headaches versus secondary headaches

Table 5 shows five-fold cross-validated performance metrics of the prediction models, with and without the ratio variables of blood cell count parameters before varying the probability

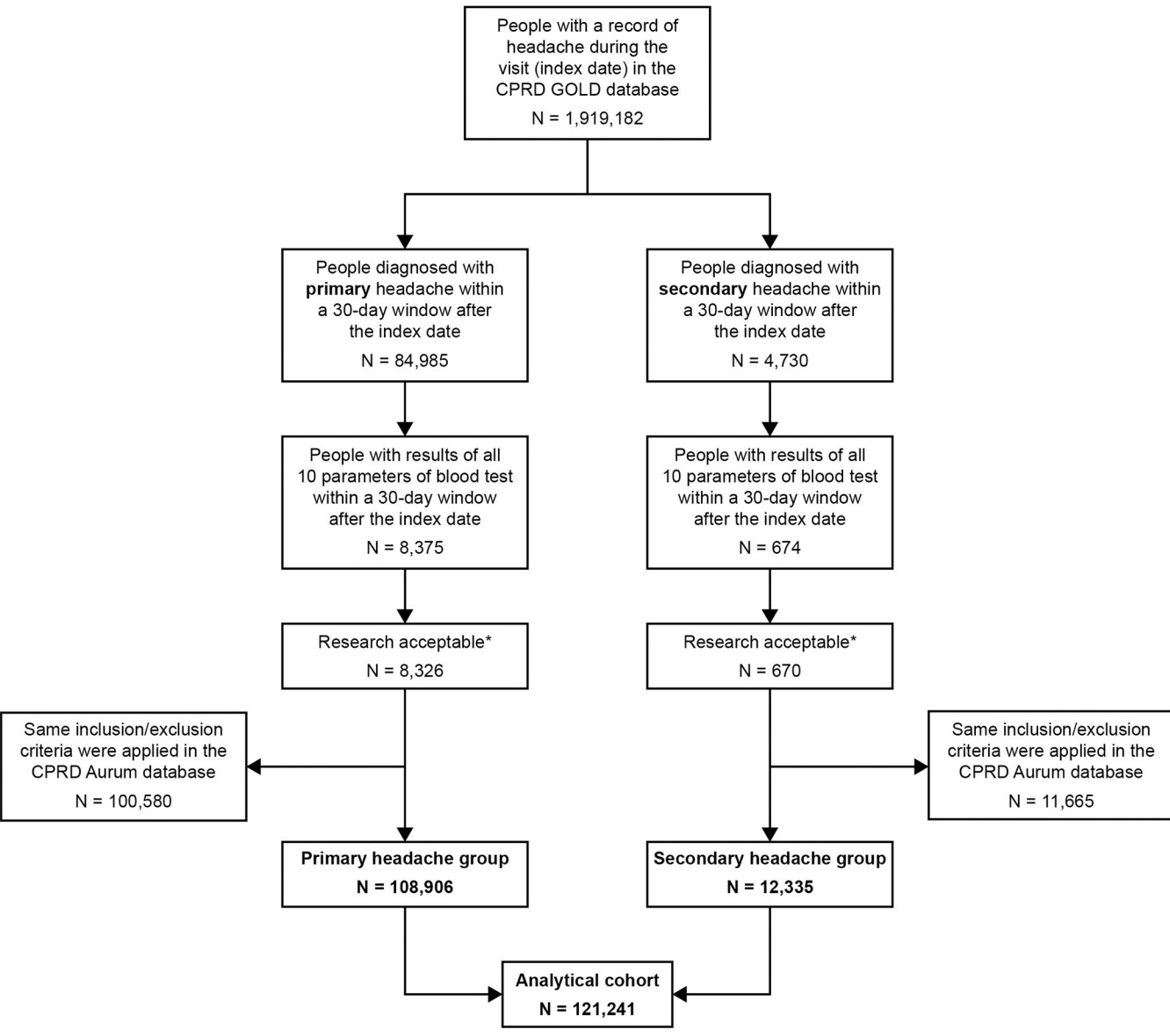

**Fig 1. Patient flow and determination of the analytical cohort.** *Patient data flagged as "research acceptable" by CPRD. Two consultation types ("follow-up/routine visit" and "mail from patient") occurring on the index date were removed from the data set. CPRD, Clinical Practice Research Datalink.

threshold, using logistic regression and random forest methods separately. The performance metrics suggested that the random forest method without the blood cell count ratio variables had an overall better predictive performance for predicting primary headaches versus secondary headaches. Spearman's correlation matrix and feature weight analysis suggested that age group, sex, total WBC count, neutrophil, and monocyte count correlated strongly with headache type (S3 Fig).

After changing the probability threshold, the final prediction model showed accurate and robust performance with a balanced accuracy at 0.7405 (Table 6), reflecting an ability to achieve a false negative rate (incorrect prediction of secondary headache as primary headache) of 10% while maintaining a false positive rate (incorrect prediction of primary headache as secondary headache) of 42% (Fig 2). The sensitivity (correct prediction of secondary headache as

**Table 1. Baseline demographic and clinical characteristics of the analytical cohort (n = 121,241).**

| | Primary headache group | | Secondary headache group | | p value[a] |
|---|---|---|---|---|---|
| | n = 108,906 | | n = 12,335 | | |
| **Age (years)** | | | | | |
| Mean, SD | 44 | 17 | 70 | 14 | < 0.001 |
| Median, IQR | 42 | 30–55 | 72 | 61–80 | |
| Min, Max | 4 | 102 | 5 | 102 | |
| **Age group, years (n, %)** | | | | | < 0.001 |
| 1–10 | 322 | 0.3 | 6 | 0.0 | |
| 11–20 | 8,436 | 7.7 | 38 | 0.3 | |
| 21–30 | 20,071 | 18.4 | 159 | 1.3 | |
| 31–40 | 21,611 | 19.8 | 313 | 2.5 | |
| 41–50 | 22,322 | 20.5 | 803 | 6.5 | |
| 51–60 | 16,596 | 15.2 | 1,588 | 12.9 | |
| 61–80 | 16,787 | 15.4 | 6,454 | 52.3 | |
| 81+ | 2,761 | 2.5 | 2,974 | 24.1 | |
| **Sex (n, %)** | | | | | < 0.001 |
| Male | 23,834 | 21.9 | 4,235 | 34.3 | |
| Female | 85,072 | 78.1 | 8,100 | 65.7 | |
| BMI (kg/m$^2$)[b] | | | | | |
| n | 85,200 | | 9,187 | | |
| Mean, SD | 27.9 | 6.2 | 28.0 | 5.7 | 0.9 |
| Median, IQR | 27.0 | 23.4–31.4 | 27.2 | 24–31.1 | |
| Min, Max | 16.6 | 52.0 | 16.7 | 51.8 | |
| **Geographic region (n, %)** | | | | | < 0.001 |
| England | 106,374 | 97.7 | 12,047 | 97.7 | |
| Northern Ireland | 608 | 0.6 | 41 | 0.3 | |
| Scotland | 637 | 0.6 | 102 | 0.8 | |
| Wales | 1,237 | 1.1 | 142 | 1.2 | |
| Unknown | 50 | 0.0 | 3 | 0.0 | |
| **Year of blood test (n, %)** | | | | | < 0.001 |
| Before 2008 | 16,258 | 14.9 | 2,214 | 17.9 | |
| 2008 and after | 92,648 | 85.1 | 10,121 | 82.1 | |
| **Month of blood test (n, %)** | | | | | 0.003 |
| Winter period (October to March) | 56,937 | 52.3 | 6,276 | 50.9 | |
| Non-winter period (April to September) | 51,969 | 47.7 | 6,059 | 49.1 | |
| Current use of medication (n, %)[c] | | | | | < 0.001 |
| Related to blood test results | 37,257 | 34.2 | 8,794 | 71.3 | |
| Not related/no medication | 71,649 | 65.8 | 3,541 | 28.7 | |

[a]For categorical variables, p values were derived from chi-squared tests; for continuous variables, p values were derived from t-tests to examine the difference between the group means for the primary and secondary headache groups.

[b]For BMI, patients with outlier results (top and bottom 1% of the values) were removed from analysis, leaving a range of values between 16.6 kg/m$^2$ and 52.0 kg/m$^2$.

[c]Current use of medication in relation to blood test results was classified according to BNF chapters: 1) Related: Antiplatelet drugs OR Corticosteroids (in Chronic Bowel Disorders) / Corticosteroids (for Respiratory Conditions) / Antileprotic Drugs / Drugs For Pneumocystis Pneumonia / Glucocorticoid Therapy / Corticosteroids And Other Immunosuppressants / Systemic Corticosteroids (in Musculoskeletal And Joint Conditions); 2) Not related: Other BNF chapters.

BMI, body mass index; BNF, British National Formulary; IQR, interquartile range; SD, standard deviation.

**Table 2. Descriptive statistics for variables from the CBC test for the analytical cohort (n = 121,241).**

| Blood test results[a] | Primary headache group n = 108,906 | | Secondary headache group n = 12,335 | | p value[b] |
|---|---|---|---|---|---|
| RBC count ($10^{12}$/L) | | | | | |
| **n** | 106,483 | | 11,796 | | |
| **Mean, SD** | 4.59 | 0.42 | 4.46 | 0.45 | < 0.001 |
| **Median, IQR** | 4.58 | 4.3–4.9 | 4.45 | 4.1–4.8 | |
| **Min, Max** | 3.47 | 5.78 | 3.47 | 5.78 | |
| Platelet count ($10^9$/L) | | | | | |
| **n** | 106,935 | | 11,642 | | |
| **Mean, SD** | 270.26 | 63.40 | 276.02 | 75.38 | < 0.001 |
| **Median, IQR** | 264.00 | 225–308 | 267.00 | 221–322 | |
| **Min, Max** | 134.00 | 510.00 | 134.00 | 510.00 | |
| MCV (fL) | | | | | |
| **n** | 106,833 | | 11,957 | | |
| **Mean, SD** | 89.10 | 5.23 | 90.45 | 5.34 | < 0.001 |
| **Median, IQR** | 89.30 | 86–92.6 | 90.70 | 87.2–94 | |
| **Min, Max** | 69.50 | 102.50 | 69.50 | 102.50 | |
| WBC count ($10^9$/L) | | | | | |
| **n** | 107,064 | | 11,657 | | |
| **Mean, SD** | 7.01 | 1.96 | 7.97 | 2.37 | < 0.001 |
| **Median, IQR** | 6.70 | 5.6–8.1 | 7.64 | 6.2–9.4 | |
| **Min, Max** | 3.40 | 14.97 | 3.40 | 14.90 | |
| Neutrophil count ($10^9$/L) | | | | | |
| **n** | 107,025 | | 11,656 | | |
| **Mean, SD** | 4.09 | 1.60 | 5.13 | 2.09 | < 0.001 |
| **Median, IQR** | 3.80 | 3–4.9 | 4.70 | 3.6–6.3 | |
| **Min, Max** | 1.40 | 11.40 | 1.40 | 11.40 | |
| Lymphocyte count ($10^9$/L) | | | | | |
| **n** | 106,957 | | 11,632 | | |
| **Mean, SD** | 2.15 | 0.66 | 1.98 | 0.74 | < 0.001 |
| **Median, IQR** | 2.09 | 1.7–2.5 | 1.90 | 1.4–2.4 | |
| **Min, Max** | 0.80 | 4.52 | 0.80 | 4.52 | |
| Monocyte count ($10^9$/L) | | | | | |
| **n** | 107,141 | | 11,767 | | |
| **Mean, SD** | 0.50 | 0.17 | 0.59 | 0.22 | < 0.001 |
| **Median, IQR** | 0.50 | 0.4–0.6 | 0.57 | 0.4–0.7 | |
| **Min, Max** | 0.20 | 1.20 | 0.20 | 1.20 | |
| Eosinophil count ($10^9$/L) | | | | | |
| **n** | 107,623 | | 12,220 | | |
| **Mean, SD** | 0.19 | 0.13 | 0.17 | 0.14 | < 0.001 |
| **Median, IQR** | 0.16 | 0.1–0.2 | 0.13 | 0.1–0.2 | |
| **Min, Max** | 0 | 0.80 | 0 | 0.80 | |
| Basophil count ($10^9$/L) | | | | | |
| **n** | 107,641 | | 12,129 | | |
| **Mean, SD** | 0.04 | 0.04 | 0.04 | 0.04 | 0.01 |
| **Median, IQR** | 0.03 | 0–0.1 | 0.03 | 0–0.1 | |
| **Min, Max** | 0 | 0.12 | 0 | 0.12 | |
| **Hemoglobin (g/dL)** | | | | | |
| **n** | 106,899 | | 11,989 | | |

(*Continued*)

**Table 2.** (Continued)

| Blood test results[a] | Primary headache group n = 108,906 | | Secondary headache group n = 12,335 | | p value[b] |
|---|---|---|---|---|---|
| Mean, SD | 13.55 | 1.29 | 13.24 | 1.42 | < 0.001 |
| Median, IQR | 13.50 | 12.7–14.4 | 13.30 | 12.3–14.2 | |
| Min, Max | 9.60 | 17.00 | 9.60 | 17.00 | |

[a]Patients with outlier results (top and bottom 1% of the values) were removed from analysis for each parameter of blood test, respectively.

[b]p values were derived from t-tests to examine the difference between the group means for the primary and secondary headache groups.

CBC, complete blood count; IQR, interquartile range; MCV, mean corpuscular volume; RBC, red blood cell; SD, standard deviation; WBC, white blood cell.

secondary headache) and specificity (correct prediction of primary headache as primary headache) of the final model were 58% and 90%, respectively (Fig 2).

## Discussion

In summary, descriptive statistical analysis revealed substantial overlap in values for the ten CBC parameters and 19 ratio variables measured in patients in both headache groups, and patients with primary and secondary headaches could not be properly differentiated based on results from the Hotelling $T^2$-test. However, we have developed a ML-based prediction model with a modest predictive accuracy to differentiate between primary and secondary headaches on the basis of readily available patient characteristics and routine blood test results.

Diagnostic procedures and acute treatment for headaches may vary across different countries, depending on factors such as catchment area, structure of the care facility, in-house protocols, and local medical staff [3]. In current clinical practice, an excess of patients presenting to the ED with a severe headache are referred for neuroimaging, despite current guidelines recommending against routine neuroimaging for headaches [33]. In one European study, neurologic examination was performed in 72.5% of patients presenting to the ED with headache; 60.9% subsequently underwent technical investigation and 53.2% had non-contrast cranial computed tomography [3]. However, unnecessary investigations should be avoided [1]; it is not appropriate, for example, to routinely use computed tomography in headache patients, because of the high cost and radiation exposure [16].

Our study suggests that a CBC-based ML algorithm could mitigate this problem, by simplifying the triage of patients who require such diagnostic procedures. Our findings indicate that age group, sex, and ten parameters that are usually collected during CBC tests represent convenient, measurable variables for use in a ML prediction model, to differentiate patients with primary and secondary headache. We also developed a prediction model with modest performance, predicting almost 60% of secondary headache patients and 90% of primary headache patients. Future studies may look at including certain clinical characteristics (such as a history of brain trauma, hypertension, epilepsy, or stroke) in the prediction model to assess whether their addition could improve its performance further.

In a similar study, the sensitivity of leukocytosis or increase in the percentage of PMNs in cases of patients with SAH was investigated, with a view to developing a non-invasive blood test to facilitate diagnosis [16]. Investigators concluded that CBC had an excellent sensitivity (89.8%) in the exclusion of SAH in non-traumatic headache. Specificity, however, was poor (46.7%), as leukocytosis can result from other headache etiologies such as migraine, temporal arteritis (giant cell arteritis), and hypertension, suggesting CBC levels could only be used to rule out, rather than confirm, SAH [16].

Our ML model has potentially important clinical implications. The reasonably low error rate (10%) of misclassifying secondary headache as primary headache could aid clinicians'

**Table 3. Descriptive statistics for blood cell ratios for the analytical cohort (n = 67,974).**

| Blood test ratios[a] | Primary headache group n = 61,887 | | Secondary headache group n = 6,087 | | p value[b] |
|---|---|---|---|---|---|
| **Platelet/RBC** | | | | | |
| Mean, SD | 60.02 | 14.96 | 62.39 | 18.09 | < 0.001 |
| Median, IQR | 58.41 | 49.3–68.9 | 59.92 | 48.8–73.2 | |
| Min, Max | 25.14 | 136.74 | 26.01 | 139.26 | |
| **WBC/RBC** | | | | | |
| Mean, SD | 1.55 | 0.43 | 1.75 | 0.50 | < 0.001 |
| Median, IQR | 1.49 | 1.2–1.8 | 1.68 | 1.4–2 | |
| Min, Max | 0.62 | 4.02 | 0.65 | 4.07 | |
| **RBC/Neutrophil** | | | | | |
| Mean, SD | 1.26 | 0.47 | 1.05 | 0.42 | < 0.001 |
| Median, IQR | 1.19 | 0.9–1.5 | 0.98 | 0.7–1.3 | |
| Min, Max | 0.31 | 3.94 | 0.35 | 3.91 | |
| **RBC/Monocyte** | | | | | |
| Mean, SD | 10.19 | 3.55 | 8.68 | 3.35 | < 0.001 |
| Median, IQR | 9.58 | 7.6–12.1 | 8.00 | 6.3–10.4 | |
| Min, Max | 2.93 | 28.60 | 2.93 | 25.35 | |
| **Monocyte/Eosinophil** | | | | | |
| Mean, SD | 3.91 | 4.28 | 5.35 | 7.65 | < 0.001 |
| Median, IQR | 2.94 | 1.9–4.7 | 3.40 | 2.1–5.8 | |
| Min, Max | 0.25 | 120.00 | 0.32 | 111.00 | |
| **Platelet/MCV** | | | | | |
| Mean, SD | 3.08 | 0.76 | 3.07 | 0.84 | 0.2 |
| Median, IQR | 2.99 | 2.5–3.5 | 2.97 | 2.5–3.6 | |
| Min, Max | 1.34 | 7.04 | 1.39 | 6.31 | |
| **Platelet/Lymphocyte** | | | | | |
| Mean, SD | 134.58 | 47.21 | 152.10 | 65.21 | < 0.001 |
| Median, IQR | 126.79 | 101.4–158.5 | 138.33 | 106.3–182.2 | |
| Min, Max | 32.00 | 542.50 | 35.24 | 547.50 | |
| **Platelet/Eosinophil** | | | | | |
| Mean, SD | 2,181.13 | 2,264.13 | 2,643.64 | 3,984.65 | < 0.001 |
| Median, IQR | 1,621.43 | 1033.3–2626.7 | 1,640.00 | 1023.3–2770 | |
| Min, Max | 167.50 | 44,100.00 | 217.33 | 48,200.00 | |
| **MCV/WBC** | | | | | |
| Mean, SD | 13.45 | 3.61 | 12.54 | 3.56 | < 0.001 |
| Median, IQR | 13.03 | 10.8–15.6 | 12.08 | 9.9–14.6 | |
| Min, Max | 5.07 | 29.47 | 5.23 | 28.74 | |
| **MCV/Neutrophil** | | | | | |
| Mean, SD | 24.55 | 9.13 | 21.28 | 8.39 | < 0.001 |
| Median, IQR | 23.00 | 18–29.4 | 19.75 | 15.3–25.6 | |
| Min, Max | 6.69 | 72.36 | 7.26 | 72.79 | |
| **Neutrophil/Lymphocyte** | | | | | |
| Mean, SD | 2.03 | 0.97 | 2.73 | 1.54 | < 0.001 |
| Median, IQR | 1.82 | 1.4–2.4 | 2.33 | 1.7–3.3 | |
| Min, Max | 0.36 | 13.11 | 0.35 | 13.38 | |
| **Neutrophil/Eosinophil** | | | | | |
| Mean, SD | 33.24 | 42.48 | 50.82 | 95.60 | < 0.001 |
| Median, IQR | 23.57 | 14.6–38.2 | 27.26 | 16.5–48 | |

*(Continued)*

**Table 3.** (Continued)

| Blood test ratios[a] | Primary headache group n = 61,887 | | Secondary headache group n = 6,087 | | p value[b] |
|---|---|---|---|---|---|
| Min, Max | 1.92 | 1,134.00 | 2.00 | 1,122.00 | |
| **Lymphocyte/Monocyte** | | | | | |
| Mean, SD | 4.73 | 1.81 | 3.81 | 1.71 | < 0.001 |
| Median, IQR | 4.45 | 3.5–5.7 | 3.50 | 2.6–4.7 | |
| Min, Max | 0.70 | 20.45 | 0.67 | 16.15 | |
| **Lymphocyte/Eosinophil** | | | | | |
| Mean, SD | 16.79 | 15.98 | 17.95 | 23.78 | < 0.001 |
| Median, IQR | 12.91 | 8.2–20.4 | 12.00 | 7.5–19.7 | |
| Min, Max | 1.33 | 428.00 | 1.38 | 444.00 | |
| **Hemoglobin/Lymphocyte** | | | | | |
| Mean, SD | 6.78 | 2.17 | 7.42 | 2.68 | < 0.001 |
| Median, IQR | 6.42 | 5.2–7.9 | 6.95 | 5.4–8.9 | |
| Min, Max | 2.36 | 20.75 | 2.42 | 20.74 | |
| **Hemoglobin/Eosinophil** | | | | | |
| Mean, SD | 109.84 | 113.15 | 127.48 | 178.17 | < 0.001 |
| Median, IQR | 80.00 | 52.6–134 | 79.41 | 52.8–135.2 | |
| Min, Max | 12.88 | 1,680.00 | 12.75 | 1,650.00 | |
| **Hemoglobin/RBC** | | | | | |
| Mean, SD | 2.96 | 0.20 | 2.99 | 0.20 | < 0.001 |
| Median, IQR | 2.98 | 2.9–3.1 | 3.00 | 2.9–3.1 | |
| Min, Max | 1.98 | 3.85 | 2.09 | 3.73 | |
| **MCV/Monocyte** | | | | | |
| Mean, SD | 198.29 | 69.50 | 175.27 | 66.77 | < 0.001 |
| Median, IQR | 185.20 | 148.2–233.4 | 162.81 | 127.1–208.1 | |
| Min, Max | 64.08 | 512.50 | 68.22 | 509.00 | |
| **MCV/Hemoglobin** | | | | | |
| Mean, SD | 6.61 | 0.63 | 6.83 | 0.72 | < 0.001 |
| Median, IQR | 6.58 | 6.2–7 | 6.78 | 6.3–7.3 | |
| Min, Max | 4.79 | 10.08 | 5.00 | 9.43 | |

[a]Only patients who had non-zero values for all ten parameters were included, so this analysis was performed on data from n = 67,974 patients. Patients with outlier results (top and bottom 1% of the values) on any parameter of the blood test and BMI were removed prior to deriving ratio variables.

[b]p values were derived from t-tests to examine the difference between the group means for the primary and secondary headache groups.

BMI, body mass index; IQR, interquartile range; MCV, mean corpuscular volume; RBC, red blood cell; SD, standard deviation; WBC, white blood cell.

**Table 4. Hypothesis testing results using the Hotelling T²-test (n = 67,974).**

| Variables in the model[a] | Patients correctly placed as having primary headaches, % | Patients correctly placed as having secondary headaches, % |
|---|---|---|
| Blood cell counts | 2 | 19 |
| Ratios of blood cell counts | 8.7 | 17 |
| Blood cell counts and ratios | 13 | 20 |

[a]Only patients who had non-zero values for all ten parameters were included, so this analysis was performed on data from n = 67,974 patients.

**Table 5. Performance metrics of the prediction models using logistic regression and random forest methods with five-fold cross-validation.**

| Performance metrics | Logistic regression | | Random forest | |
|---|---|---|---|---|
| | Without blood cell count ratios | With blood cell count ratios | Without blood cell count ratios | With blood cell count ratios |
| Accuracy, mean ± SD | 0.9143 ± 0.0032 | 0.9103 ± 0.0014 | 0.9181 ± 0.0021 | 0.9174 ± 0.0023 |
| Balanced accuracy, mean ± SD | 0.5704 ± 0.0076 | 0.5033 ± 0.0013 | 0.5905 ± 0.0019 | 0.5749 ± 0.0012 |
| Average precision, mean ± SD | 0.4131 ± 0.0208 | 0.1379 ± 0.0084 | 0.4501 ± 0.0136 | 0.4382 ± 0.0106 |
| F1-score, mean ± SD | 0.2403 ± 0.0214 | 0.0146 ± 0.0049 | 0.2939 ± 0.0125 | 0.2563 ± 0.0079 |
| AUC, mean ± SD | 0.8624 ± 0.0079 | 0.5818 ± 0.0140 | 0.8586 ± 0.0069 | 0.8529 ± 0.0075 |

All models included the features: age group, sex, and laboratory blood test results of ten CBC parameters, with five-fold cross-validation and without any model optimization focusing on the prediction of secondary headaches.

AUC, area under the receiver operator characteristic curve; CBC, complete blood count; SD, standard deviation.

decision-making on which patients to refer for further examination (e.g. neuroimaging and/or lumbar puncture), thereby avoiding unnecessary procedures and reducing the drain on healthcare resources. It is envisaged that our model could be used alongside taking a detailed headache history and, if indicated, a thorough neurologic examination. In cases of abnormal findings, neuroimaging should be performed to rule-out secondary headaches. Currently, if there is a clinical suspicion of SAH, computed tomography is undertaken, followed by lumbar puncture if the scan is inconclusive [34]. As our model is able to distinguish between primary and secondary headaches with a sensitivity of 58%, a specificity of 90%, a false negative rate of 10% and false positive rate of 42%, it is hoped that it may help reduce the number of unnecessary procedures. Furthermore, the model will be of particular use in countries where neuroimaging is not readily available, but should be used with caution, particularly when headache history is sparse.

As a point to consider, because primary headache accounted for approximately 90% of all headache cases in this study, a default prediction model with good prediction accuracy may have been informed by an imbalanced data set (i.e. a model that predicts primary headache for all patients will be correct in 90% of cases). However, the model developed herein accounted for such imbalanced data and focused on the accurate prediction of secondary headaches, resulting in a modest predictive model with a balanced accuracy of 0.7405. Furthermore, as healthcare progresses, in the future our model could be expanded to include other relevant parameters to further improve the model performance.

When comparing random forest and logistic regression methods, the former marginally outperformed the latter in nearly all prediction performance metrics, particularly those suitable for imbalanced data sets (balanced accuracy, average precision, and F1-score). This is

**Table 6. Performance metrics of the final prediction model on the test set using the random forest method.**

| Performance measure | Without blood cell count ratios | With blood cell count ratios |
|---|---|---|
| Accuracy | 0.6067 | 0.5546 |
| Balanced accuracy | 0.7405 | 0.7137 |
| Average precision | 0.4375 | 0.4239 |
| F1-score | 0.2932 | 0.2689 |
| AUC | 0.8519 | 0.8417 |

The model included the features: age group, sex and laboratory blood test results of ten CBC parameters, with model optimization focusing on the prediction of secondary headaches. The model was evaluated using a single 80:20 train/test dataset split.

AUC, area under the receiver operator characteristic curve; CBC, complete blood count.

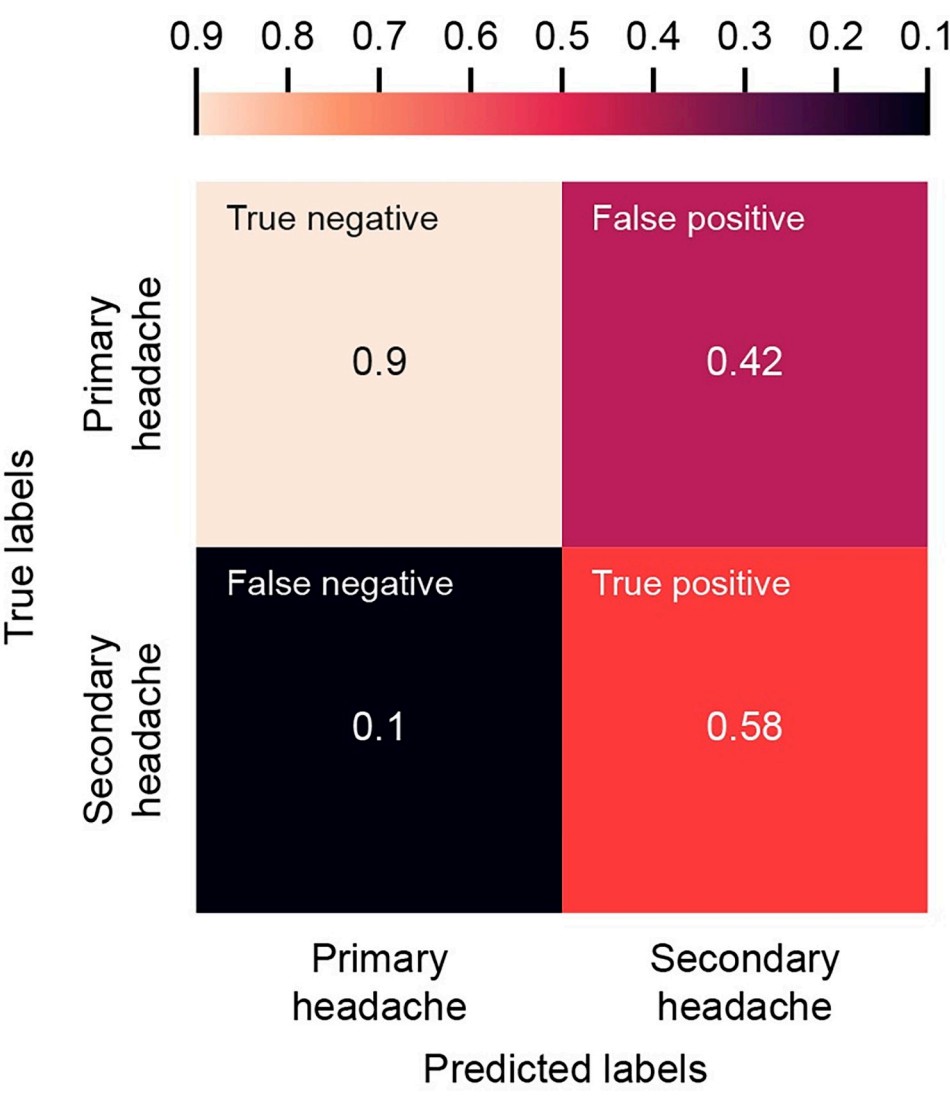

**Fig 2. Confusion matrix (normalized to ratio instead of patient count) of the final prediction model on the test set using the random forest method.** Features included in the model are: age group, sex, and laboratory blood test results of ten CBC parameters. CBC, complete blood count.

likely due to the fact that the random forest model is fundamentally a large number of uncorrelated individual decision trees operating as an ensemble/committee and is hence better suited to capture interactions between variables. The logistic regression model on the other hand, has a linear function and cannot capture such interactions.

Correlated features have an influence on the prediction performance of the model, which was suggested by performance metrics to be slightly compromised by the features of blood cell ratios. Although this is counterintuitive, according to the premise that "more data is better", data are required to be independent and identically distributed, and it follows that such correlations are detrimental for the performance of ML techniques. The features of blood cell ratios are highly inter-dependent, being strongly correlated both to each other and to the original features of blood cell counts, thus violating this data premise.

We attempted to simplify the model, whilst maintaining good performance, by using feature selection techniques, including lasso regularization and recursive feature elimination. As

correlation and feature weight analyses indicated that age group, total WBC count, monocyte count, and neutrophil count were of the greatest significance for the model, only these features were included. However, none of the ML feature selection techniques yielded a better result compared with the final model that included age group, sex, and laboratory results of ten parameters usually collected during a CBC test. This is probably due to the final prediction model demonstrating only a marginal improvement when compared with the ML selection techniques focusing on a certain metric, e.g. accuracy. In addition, many simpler models with fewer features resort to the baseline "guess" model of predicting every patient as primary headache regardless of the data.

The key strength of this study is the large sample size of >120,000 patients and the use of RWD from the UK. Awareness in the UK of the difficulties around headache diagnosis and treatment has prompted the training of GPs with special interest in headache [10] and the establishment of a network of headache centers [11]; our findings could potentially inform these endeavors.

The use of primary care data rather than data directly from ED settings is a limitation of the study, due to the extrapolation performed, which makes it difficult to assess the real-life advantages of the ML approach to differentiating headache types in the ED environment. Further studies using ED data are warranted to validate the algorithm described here for this differentiation. This study considered primary and secondary headaches as two groups of heterogeneous conditions; future work could evaluate diagnostic accuracy of measurements from CBC tests in different forms of primary and secondary headache. As this was a retrospective study of anonymized data some patients included in the study may, for example, have comorbidities, such as an underlying infection or inflammatory condition, or may be using medications that alter certain CBC markers. Further research should be carried out in specific patient populations, such as immunocompromised individuals, to elucidate the potential prognostic value of CBC and CBC-derived ratio parameters in differentiating primary and secondary headaches.

In conclusion, this study demonstrated the use of a ML approach to create a prediction model with a modest level of performance to differentiate patients with primary and secondary headache in clinical settings.

## Supporting information

**S1 Table. List of read codes used to identify diagnosis of primary headache disorders in CPRD GOLD. NOS, not otherwise specified.**
(DOCX)

**S2 Table. List of medical and read codes used to identify diagnosis of primary headache disorders in CPRD Aurum. NOS, not otherwise specified.**
(DOCX)

**S3 Table. List of read codes used to identify diagnosis of secondary headache disorders in CPRD GOLD. NOS, not otherwise specified.**
(DOCX)

**S4 Table. List of medical and read codes used to identify diagnosis of secondary headache disorders in CPRD Aurum.** CVA, cerebrovascular accident; NOS, not otherwise specified.
(DOCX)

**S5 Table. List of read codes used to identify headache symptoms in CPRD GOLD. C/O, complains of; NOS, not otherwise specified.**
(DOCX)

**S6 Table. List of medical and read codes used to identify headache symptoms in CPRD Aurum.** C/O, patient complains of; NOS, not otherwise specified.
(DOCX)

**S7 Table. List of codes used to define laboratory blood test type in CPRD GOLD.**
(DOCX)

**S8 Table. List of codes used to define laboratory blood test type in CPRD Aurum.**
(DOCX)

**S9 Table. List of codes used to define consultation type in CPRD GOLD.**
(DOCX)

**S10 Table. List of codes used to define consultation type in CPRD Aurum.**
(DOCX)

**S1 Fig.** Distribution of the following 10 parameters from CBC test results by headache group (data cleaned by removing the extreme values of blood test results): A. RBC count ($10^{12}$/L), B. platelet count ($10^9$/L), C. MCV (fL), D. WBC count ($10^9$/L), E. neutrophil count ($10^9$/L), F. lymphocyte count ($10^9$/L), G. monocyte count ($10^9$/L), H. eosinophil count ($10^9$/L), I. basophil count ($10^9$/L), J. hemoglobin (g/dL). CBC, complete blood count; MCV, mean corpuscular volume; RBC, red blood cell; WBC, white blood cell.
(DOCX)

**S2 Fig.** Distribution by headache group of the following ratios: A. platelet/RBC, B. WBC/RBC, C. RBC/neutrophil, D. RBC/monocyte, E. monocyte/eosinophil, F. platelet/MCV, G. platelet/lymphocyte, H. platelet/eosinophil, I. MCV/WBC, J. MCV/neutrophil, K. neutrophil/lymphocyte, L. neutrophil/eosinophil, M. lymphocyte/monocyte, N. lymphocyte/eosinophil, O. hemoglobin/lymphocyte, P. hemoglobin/eosinophil, Q. hemoglobin/RBC, R. MCV/monocyte and S. MCV/hemoglobin. Solid vertical lines indicate the mean of the distribution. MCV, mean corpuscular volume; RBC, red blood cell; WBC, white blood cell.
(DOCX)

**S3 Fig.** A. Spearman's correlation matrix and B. feature weight analysis for the features age group, sex and 10 variables from the CBC test, derived from the logistic regression model. CBC, complete blood count; MCV, mean corpuscular volume; RBC, red blood cell; WBC, white blood cell.
(DOCX)

## Acknowledgments

The authors gratefully thank Marie Stobbe (Roche Diagnostics) for project coordination and support, Iori Namekawa (F. Hoffmann-La Roche Ltd.) for data management, Martine Kallemeijn (Roche Diagnostics) for critical review of the manuscript, and Simon John Davidson (Freelance Hemostasis Consultant, London, UK) for his input into study design and conception. Third-party medical writing assistance, under the direction of the authors, was provided by Anna King, PhD and Elizabeth Hilsley, BSc, of Ashfield MedComms (Macclesfield, UK), an Inizio company.

## Author Contributions

**Conceptualization:** Fei Yang, Chuang Liu, Emilie Dejean.

**Data curation:** Fei Yang.

**Formal analysis:** Fei Yang, Tong Meng, Ben Torben-Nielsen, Carsten Magnus, Emilie Dejean.

**Methodology:** Fei Yang.

**Writing – original draft:** Fei Yang, Ben Torben-Nielsen, Carsten Magnus, Chuang Liu, Emilie Dejean.

**Writing – review & editing:** Fei Yang, Tong Meng, Ben Torben-Nielsen, Carsten Magnus, Chuang Liu, Emilie Dejean.

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
