## [Decision Letter · Decision Letter 0]

28 Oct 2022

PONE-D-22-27401A machine learning approach to support triaging of primary versus secondary headache patients using complete blood countPLOS ONE

Dear Dr. Yang,

Thank you for submitting your manuscript to PLOS ONE. After careful consideration, we feel that it has merit but does not fully meet PLOS ONE’s publication criteria as it currently stands. Therefore, we invite you to submit a revised version of the manuscript that addresses the points raised during the review process.

In particular, Reviewer 1 raised important concerns on the rationale for applying the proposed model for the triaging of headache patients. Moreover, further acknowledgment and discussion of the study limitations is strongly recommended.

We look forward to receiving your revised manuscript.

Kind regards,

Jacopo Sabbatinelli, MD, PhD

Academic Editor

PLOS ONE

Journal Requirements:

2. Please note that PLOS ONE has specific guidelines on code sharing for submissions in which author-generated code underpins the findings in the manuscript. In these cases, all author-generated code must be made available without restrictions upon publication of the work. Please review our guidelines at https://journals.plos.org/plosone/s/materials-and-software-sharing#loc-sharing-code and ensure that your code is shared in a way that follows best practice and facilitates reproducibility and reuse. New software must comply with the Open Source Definition.

"I have read the journal's policy and the authors of this manuscript have the following

competing interests: 

FY, TM, BT-N and CM are employees of Roche. CL and ED were employees of Roche at the time the study was conducted." 

Reviewers' comments:

Reviewer's Responses to Questions

**Comments to the Author**

1. Is the manuscript technically sound, and do the data support the conclusions?

Reviewer #1: No

Reviewer #2: Yes

2. Has the statistical analysis been performed appropriately and rigorously? 

Reviewer #1: Yes

Reviewer #2: Yes

3. Have the authors made all data underlying the findings in their manuscript fully available?

Reviewer #1: Yes

Reviewer #2: Yes

4. Is the manuscript presented in an intelligible fashion and written in standard English?

Reviewer #1: No

Reviewer #2: Yes

5. Review Comments to the Author

Reviewer #1: This study aimed to investigate the accuracy of routine blood tests in discriminating patients with primary and secondary headaches. Real-world data from 121,241 patients attending a GP practice were collected. The predictive value of 10 standard measurements of complete blood count (CBC) test, ratios of the 10 CBC test parameters, and patients’ demographic characteristics was assessed using two machine learning models. The final predictive model showed modest predictive accuracy. The sensitivity, specificity, false negative rate and false positive rate were 58%, 90%, 10% and 61 42%, respectively.

The mayor limit of the study is the lack of evidence supporting the utilities of using the selected measurements of complete blood count (CBC) in discriminating primary and secondary headaches. Authors should explain why the different CBC measures could be relevant in the differential diagnosis between primary and secondary headaches. Another important limitation is that the Introduction, Discussion and the aim of the study seem to be focused on the role of CBC measures in headache diagnosis in the ED, while the study at the end was based on patients presenting to a GP practice.

The following points should also be revised:

• Keywords: some of the keywords reported are repetitive. Please reduce the number of key words.

• In the introduction, the authors first stated that the NLR ration was described to be higher in migraine patients. Then, they reported other studies showing an association between NLR ration and secondary headaches. This point is quite confusing and should be clarified.

• In the Introduction, when authors explained the aim of the study, they should not refer to emergency physicians but they should highlight the utility to support GP in the differential diagnosis between primary and secondary headaches.

• Authors should also report which type of primary and secondary headache had the patients included in the study.

• Did authors investigate the diagnostic accuracy of CBC measures based on the different forms of primary and secondary headaches? It would be interesting to see if the accuracy change accorign to the different types of headaches.

Reviewer #2: This article is very interesting, in its attempt to diagnosis primary vs secondary headache with various CBC analysis through machine learning techniques.

The authors have addressed important limitations. CBC analysis is an oversimplified means, even is some causes of immunocompromised individuals. More discussion of limitations is warranted.

In addition, authors should discuss added value of adding clinical characteristics such as the presence of red flags, to increase the prediction.

At what point in the algorithm should be CBC be used to guide diagnosis?

The should be some caution added to the potential for over relevance on CBC- as headache history is often sparce.

Gold standards of SAH diagnosis should be communicated, and again how will CBC add value? What situations? Low income countries were imaging is not readily available?

6. PLOS authors have the option to publish the peer review history of their article (what does this mean?). If published, this will include your full peer review and any attached files.

Reviewer #1: No

Reviewer #2: No

---

## [Author Response · Author response to Decision Letter 0]

24 Jan 2023

Author response: We believe the manuscript is now styled according to PLOS ONE requirements, but please do not hesitate to let us know if any further modifications are needed. All style-type changes have been marked up throughout the manuscript. 

2. Please note that PLOS ONE has specific guidelines on code sharing for submissions in which author-generated code underpins the findings in the manuscript. In

these cases, all author-generated code must be made available without restrictions upon publication of the work. Please review our guidelines at

https://journals.plos.org/plosone/s/materials-and-software-sharing#loc-sharing-code

 and ensure that your code is shared in a way that follows best practice and

facilitates reproducibility and reuse. New software must comply with the Open Source Definition.

Author response: We thank the journal for bringing this to our attention. We are not in a position for the code used in the machine-learning model reported in this manuscript to be made freely available; however, we would make the code available to qualified researchers on request.

In view of this, we kindly ask that the data availability statement is updated as follows (new text underlined): 

“This study is based in part on data from the Clinical Practice Research Datalink obtained under license from the UK Medicines and Healthcare products Regulatory Agency. The data are provided by patients and collected by the NHS as part of their care and support. The interpretation and conclusions contained in this study are those of the authors alone. The code used in the machine-learning model reported in this manuscript is available to qualified researchers on request.”

"I have read the journal's policy and the authors of this manuscript have the following

competing interests:

FY, TM, BT-N and CM are employees of Roche. CL and ED were employees of Roche at the time the study was conducted."

Please confirm that this does not alter your adherence to all PLOS ONE policies on sharing data and materials, by including the following statement: "This does not

alter our adherence to PLOS ONE policies on sharing data and materials.” (as detailed online in our guide for authors http://journals.plos.org/plosone/s/competinginterests). If there are restrictions on sharing of data and/or materials, please state these. Please note that we cannot proceed with consideration of your article until this information has been declared. Please include your updated Competing Interests statement in your cover letter; we will change the online submission form on your behalf. 

Author response: We confirm that the Competing Interests statement submitted with the original manuscript is still accurate for all authors on this manuscript. The Competing Interests statement does not alter our adherence to PLOS ONE policies on sharing data and materials.

Reviewer #1 Comments:

1. This study aimed to investigate the accuracy of routine blood tests in discriminating patients with primary and secondary headaches. Real-world data from 121,241 patients attending a GP practice were collected. The predictive value of 10 standard measurements of complete blood count (CBC) test, ratios of the 10 CBC test parameters, and patients’ demographic characteristics was assessed using two machine learning models. The final predictive model showed modest predictive accuracy. The sensitivity, specificity, false negative rate and false positive rate were 58%, 90%, 10% and 61 42%, respectively.

Author response: We thank Reviewer 1 for reviewing this manuscript and providing their comments, which we have addressed below. 

The major limit of the study is the lack of evidence supporting the utilities of using the selected measurements of complete blood count (CBC) in discriminating primary and secondary headaches. Authors should explain why the different CBC measures could be relevant in the differential diagnosis between primary and secondary headaches.

Author response: To address this comment, the introduction has been modified and text has been added to introduce a study by Kilic et al. (Pages 6–8, Lines 110–139). This study was originally included in the discussion only, but it is now mentioned in both the introduction and the discussion. As a result, some text referring to this study in the discussion has been deleted to avoid repetition (Page 30, Lines 363–375). A further study by Kamisli et al. has also been added to the introduction. These citations have been added to the reference list and the existing references renumbered accordingly.

(Pages 6–8, Lines 110–139): “A complete blood count (CBC) is one of the most commonly requested blood tests [15] and its results may provide important diagnostic and prognostic biomarkers indicating underlying causes of headache. Several CBC parameters have been investigated for their ability to distinguish between primary and secondary headaches. A retrospective study of patients presenting to the ED with headache reported that leukocytosis or an increase in the percentage of polymorphonuclear leukocytes (PMNs) had a sensitivity of 89.8%, a specificity of 46.7%, a positive predictive value of 82.1%, and a negative predictive value of 62.8% for diagnosing SAH within 6 or 12 hours of ED admission [16]. The ratio of neutrophils to lymphocytes (NLR) has also received increasing attention as a diagnostic and prognostic marker of inflammation and can be easily calculated from standard measurements of CBC tests. A retrospective study found NLR to be higher in people presenting to the ED with a migraine attack versus people without a headache [17]. A further retrospective, single-center analysis in ED patients presenting with headache accompanied by nausea and vomiting, found that NLR could distinguish between those with migraine and those with SAH [16]. Although this retrospective study involved a limited number of clinical cases, median NLR values were found to be significantly higher in patients with SAH compared with those with migraine and other headaches (both p < 0.001) [18]. 

Measurements derived from CBC tests have also been evaluated in the prediction of secondary headache severity. In one retrospective, single-center review, patients with SAH whose leukocyte count was >15 x 109/L during admission were more than three times more likely to develop vasospasm [19]. In another study of patients with SAH, the authors reported that patients admitted with spontaneous SAH with a leukocyte count of >20,000 had a mortality rate of 50% [20]. Furthermore, mean platelet volume and platelet distribution width has been shown to be increased in patients with cerebral venous thrombosis and brain parenchymal lesions compared with patients with cerebral venous thrombosis without lesions [21].

(Page 30, Lines 363–375): “In a similar study, the sensitivity of leukocytosis or increase in the percentage of polymorphonuclear leukocytes PMNs in cases of patients with SAH was investigated, with a view to developing a non-invasive blood test to facilitate diagnosis [16]. Patients with confirmed, non-traumatic SAH, exhibited leukocytosis and an elevated PMN percentage within 6 or 12 hours of ED admission. Sensitivity, specificity, negative predictive value and positive predictive value of leukocytosis or elevated PMN percentage in the diagnosis of SAH was 89.8% (95% confidence intervals [CI] 84.5–93.5), 46.7% (95% CI 39.6–53.9), 82.1% (95% CI 73.5–88.4) and 62.8% (95% CI 56.8–68.4), respectively, on initial admission. Investigators concluded that CBC had an excellent sensitivity (89.8%) in the exclusion of SAH in non-traumatic headache. Specificity, however, was poor (46.7%), as leukocytosis can result from other headache etiologies such as migraine, temporal arteritis (giant cell arteritis), and hypertension, suggesting CBC levels could only be used to rule out, rather than confirm, SAH [16].

2. Another important limitation is that the Introduction, Discussion and the aim of the study seem to be focused on the role of CBC measures in headache diagnosis in the ED, while the study at the end was based on patients presenting to a GP practice

Author response: In the manuscript, we refer to studies reporting data from ED settings because this is where the diagnosis of severe headache is particularly applicable, as these patients typically present to the ED. We agree that further studies using ED data are needed and would like to direct the reviewer to where we have explained the reasons for using primary care data in the study design section (Page 9, Lines 148–150) and highlighted this as a limitation in the discussion section (Pages 32–33, Lines 428–432). 

(Page 9, Lines 148–150): “Due to the limitation of accessing RWD in the ED and evaluating the suitability to the study objectives, primary care data from the UK Clinical Practice Research Datalink (CPRD) were used.”

(Pages 32–33, Lines 428–432): “The use of primary care data rather than data directly from ED settings is a limitation of the study, due to the extrapolation performed, which makes it difficult to assess the real-life advantages of the ML approach to differentiating headache types in the ED environment. Further studies using ED data are warranted to validate the algorithm described here for this differentiation.”

To address the reviewer’s comment and improve the clarity of the manuscript, we have also amended the following text:

(Page 5, Lines 83–84): “Thus, accurate diagnosis of the underlying cause of headache and treatment initiation in the ED can be particularly critical.”

(Page 6, Lines 99–101): “This can, together with patient anxiety and legal concerns, result in multiple appointments ED visits and unnecessary investigations, which may subsequently increase diagnosis times, healthcare costs, and economic burden [1, 4, 13].”

(Page 6, Lines 102–103): “The effective triaging of patients presenting with primary versus secondary headache , particularly in the ED, is an important and currently unmet need [1, 5, 14].” 

(Page 8, Lines 140–144): “Aligning with the concept of using routine blood test results as a triaging tool to assist emergency physicians when deciding whether to perform neuroimaging on patients presenting with severe headache, this study was designed to evaluate a machine learning (ML)-based approach to classify primary versus secondary headache using real-world data (RWD) derived from a large patient group.” 

(Page 29, Lines 343–345): “Diagnostic procedures and acute treatment for headaches may vary across EDs in different countries, depending on factors such as catchment area, structure of the care facility of their particular ED, in-house protocols and local medical staff [3].” 

3. The following points should also be revised:

a) Keywords: some of the keywords reported are repetitive. Please reduce the number of key words.

Author response: To address this comment, the keywords have been updated as follows (Page 2, Lines 24–26): 

“Key words: headache; headache disorders; headache triaging; primary headache; secondary headache; machine learning; complete blood count; neurology; CPRD; real-world data; real-world evidence; primary care; machine learning prediction”

b) In the introduction, the authors first stated that the NLR ration was described to be higher in migraine patients. Then, they reported other studies showing an association between NLR ration and secondary headaches. This point is quite confusing and should be clarified.

Author response: The first reference (Karabulut et al. 2016) compares people with migraines to people without headaches, whereas the second reference (Eryigit et al. 2017) compares people with SAH to people with migraines. To address this comment, we have amended the following text for clarity (Page 7, Lines 122–127): 

Please note that deleted text has been omitted in the excerpt from the manuscript below for clarity.

”A retrospective study found NLR to be higher in people presenting to the ED with a migraine attack versus people without a headache [17]. A further retrospective, single-center analysis in ED patients presenting with headache accompanied by nausea and vomiting, found that NLR could distinguish between those with migraine and those with SAH [16].”

From this, it could be extrapolated that people without a headache have a low NLR, people experiencing a migraine attack have a medium NLR and people experiencing SAH have a high NLR; however, as NLR values cannot be compared between distinct studies, we have not included this interpretation in the manuscript. 

c) In the Introduction, when authors explained the aim of the study, they should not refer to emergency physicians but they should highlight the utility to support GP in the differential diagnosis between primary and secondary headaches.

Author response: We would like to direct the reviewer to our response to Comment #2 above, where we have addressed this.

d) Authors should also report which type of primary and secondary headache had the patients included in the study.

e) Did authors investigate the diagnostic accuracy of CBC measures based on the different forms of primary and secondary headaches? It would be interesting to see if the accuracy change according to the different types of headaches.

Author response: We did not investigate the diagnostic accuracy of CBC measures for different forms of primary and secondary headaches. We acknowledge this very interesting research question but suggest that it falls outside the scope of the current study, which considered primary and secondary headaches as two groups of heterogeneous conditions. To address comments 3d) and 3e), the following text has been added to the discussion section (Page 33, Lines 432–434):

“This study considered primary and secondary headaches as two groups of heterogeneous conditions; future work could evaluate diagnostic accuracy of measurements from CBC tests in different forms of primary and secondary headache.”

We would also like to direct the reviewer to the “Outcome definitions and study variables” sub-section of the materials and methods section, together with Tables S1–S6, which provide an overview of headache types included in the study (Page 10, Lines 177–182):

“Primary headaches were defined as migraine, tension-type headache, and cluster headache, while secondary headaches were defined as those caused by ischemic stroke, cerebral venous thrombosis, hemorrhage (including SAH), arteritis, and angiitis (S1–S6 Tables). In the event that a patient had diagnoses contributing to both primary and secondary headaches during the same episode, the patient was categorized as having secondary headache.”

Reviewer #2 Comments:

This article is very interesting, in its attempt to diagnosis primary vs secondary headache with various CBC analysis through machine learning techniques. The authors have addressed important limitations. 

Author response: We thank Reviewer 2 for reviewing this manuscript and providing their comments, which we have addressed below. 

1. CBC analysis is an oversimplified means, even is some causes of immunocompromised individuals. More discussion of limitations is warranted.

Author response: To address this comment, the following text has been added to the discussion section (Page 33, Lines 434–440): 

“As this was a retrospective study of anonymized data some patients included in the study may, for example, have comorbidities, such as an underlying infection or inflammatory condition, or may be using medications that alter certain CBC markers. Further research should be carried out in specific patient populations, such as immunocompromised individuals, to elucidate the potential prognostic value of CBC and CBC-derived ratio parameters in differentiating primary and secondary headaches.”

2. In addition, authors should discuss added value of adding clinical characteristics such as the presence of red flags, to increase the prediction. 

Author response: To address this comment, the following text has been added to the discussion section (Page 30, Lines 360–362): 

“Future studies may look at including certain clinical characteristics (such as a history of brain trauma, hypertension, epilepsy, or stroke) in the prediction model to assess whether their addition could improve its performance further.”

3. At what point in the algorithm should CBC be used to guide diagnosis?

4. There should be some caution added to the potential for over relevance on CBC- as headache history is often sparce.

5. Gold standards of SAH diagnosis should be communicated, and again how will CBC add value? What situations? Low income countries were imaging is not readily available?

Author response: To address comments 3–5, a reference to a study by Marcolini and Hine has been added and the following text has been included in the discussion section (Pages 30–31, Lines 380–389): 

“It is envisaged that our model could be used alongside taking a detailed headache history and, if indicated, a thorough neurologic examination. In cases of abnormal findings, neuroimaging should be performed to rule-out secondary headaches. Currently, if there is a clinical suspicion of SAH, computed tomography is undertaken, followed by lumbar puncture if the-scan is inconclusive [34]. As our model is able to distinguish between primary and secondary headaches with a sensitivity of 58%, a specificity of 90%, a false negative rate of 10% and false positive rate of 42%, it is hoped that it may help reduce the number of unnecessary procedures. Furthermore, the model will be of particular use in countries where neuroimaging is not readily available, but should be used with caution, particularly when headache history is sparse.”

24 January 2023

Data availability statement queries

1. We note your current Data Availability statement is:

"No - some restrictions will apply"

"This study is based in part on data from the Clinical Practice Research Datalink obtained under license from the UK Medicines and Healthcare products Regulatory Agency. The data are provided by patients and collected by the NHS as part of their care and support. The interpretation and conclusions contained in this study are those of the authors alone. The code used in the machine-learning model reported in this manuscript is available to qualified researchers on request. The data underlying the results presented in the study are available from the CPRD (https://cprd.com/)."

Before we can proceed, please address the following prompts:

a.) Are these third party data (i.e., data not owned or collected by the author(s))? If these are indeed third party data, please confirm if others may access the relevant data sets at the provided URL, https://cprd.com/. If not, please explain how others can access these datasets and confirm that others would be able to access these data in the same manner as the authors. Please also confirm that the authors did not have any special access privileges that others would not have.

b.) If these are not third party data but there are ethical or legal restrictions on sharing a de-identified data set, please explain them in detail (e.g., data contain potentially identifying or sensitive patient information, etc.) and who has imposed them (e.g., a Research Ethics Committee or Institutional Review Board, etc.). Please also provide non-author contact information* for a data access committee, ethics committee, or other institutional body to which data requests may be sent.

Author response: The data used in this study are third-party data (i.e., data not owned or collected by the author(s)) provided by the Clinical Practice Research Datalink (CPRD) under a license with the UK Medicines and Healthcare products Regulatory Agency (MHRA). A sub-license agreement/third-party agreement is required for data access by any third parties (e.g., journal editors, reviewers, etc.) who were not included in the research team of the originally approved study protocol by the CPRD's Research Data Governance (RDG) process. We confirm that the authors did not have any special access privileges that others would not have.

Others can also apply for accessing the raw data (i.e., the data used to generate relevant datasets for the current study) at the provided URL, https://cprd.com/data-access/.

Please also kindly note, the following statement must be included per relevant CPRD policies:

“This study is based in part on data from the Clinical Practice Research Database obtained under license from the UK Medicines and Healthcare products Regulatory Agency. However, the interpretation and conclusions contained in this report are those of the author(s) alone”.

---

## [Decision Letter · Decision Letter 1]

10 Feb 2023

A machine learning approach to support triaging of primary versus secondary headache patients using complete blood count

PONE-D-22-27401R1

Dear Dr. Yang,

We’re pleased to inform you that your manuscript has been judged scientifically suitable for publication and will be formally accepted for publication once it meets all outstanding technical requirements.

Kind regards,

Jacopo Sabbatinelli, MD, PhD

Academic Editor

PLOS ONE

Additional Editor Comments (optional):

Reviewers' comments:

Reviewer's Responses to Questions

**Comments to the Author**

1. If the authors have adequately addressed your comments raised in a previous round of review and you feel that this manuscript is now acceptable for publication, you may indicate that here to bypass the “Comments to the Author” section, enter your conflict of interest statement in the “Confidential to Editor” section, and submit your "Accept" recommendation.

Reviewer #1: All comments have been addressed

2. Is the manuscript technically sound, and do the data support the conclusions?

Reviewer #1: Yes

3. Has the statistical analysis been performed appropriately and rigorously? 

Reviewer #1: Yes

4. Have the authors made all data underlying the findings in their manuscript fully available?

Reviewer #1: Yes

5. Is the manuscript presented in an intelligible fashion and written in standard English?

Reviewer #1: Yes

6. Review Comments to the Author

Reviewer #1: The authors have satisfactorily responded to all my questions and made the necessary changes to the manuscript which is much improved.

7. PLOS authors have the option to publish the peer review history of their article (what does this mean?). If published, this will include your full peer review and any attached files.

Reviewer #1: No

---

## [Editor Report · Acceptance letter]

24 Feb 2023

PONE-D-22-27401R1 

A machine learning approach to support triaging of
primary versus secondary headache patients using
complete blood count 

Dear Dr. Yang:

I'm pleased to inform you that your manuscript has been deemed suitable for publication in PLOS ONE. Congratulations! Your manuscript is now with our production department. 

Kind regards, 

on behalf of

Dr. Jacopo Sabbatinelli 

Academic Editor

PLOS ONE